# The Potential of Fibroblast Transdifferentiation to Neuron Using Hydrogels

Fahsai Kantawong

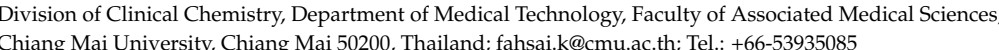

Division of Clinical Chemistry, Department of Medical Technology, Faculty of Associated Medical Sciences, Chiang Mai University, Chiang Mai 50200, Thailand; fahsai.k@cmu.ac.th; Tel.: +66-53935085

**Abstract:** Currently there is a big drive to generate neurons from differentiated cells which would be of great benefit for regenerative medicine, tissue engineering and drug screening. Most studies used transcription factors, epigenetic reprogramming and/or chromatin remodeling drugs which might reflect incomplete reprogramming or progressive deregulation of the new program. In this review, we present a potential different method for cellular reprogramming/transdifferentiation to potentially enhance regeneration of neurons. We focus on the use of biomaterials, specifically hydrogels, to act as non-invasive tools to direct transdifferentiation, and we draw parallel with existing transcriptional and epigenetic methods. Hydrogels are attractive materials because the properties of hydrogels can be modified, and various natural and synthetic substances can be employed. Incorporation of extracellular matrix (ECM) substances and composite materials allows mechanical properties and degradation rate to be controlled. Moreover, hydrogels in combinations with other physical and mechanical stimuli such as electric current, shear stress and tensile force will be mentioned in this review.

**Keywords:** hydrogels; transdifferentiation; reprogramming; epigenetics; chromatin remodeling

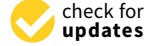

## 1. Introduction

Nerve regeneration is a relatively slow process. Damage to the nervous system leads to local problems depending on the organ and cause of that damage. The in vitro generation of neurons from embryonic stem cells (ES) is a promising approach to produce cells suitable for neural tissue repair and cell-based replacement therapies of the nervous system. Embryonic stem cell research has been proven to work in animal models but the stem cells are entrenched in ethical debate. Mesenchymal stem cells/marrow stromal cells (MSCs) have been extensively tested and proven effective to treat neurodegenerative diseases but there is a very small number of stem cells in adult human, and research is not in agreement as to whether the cells can truly transdifferentiate or if they perform a supporting role (e.g., opsonization, immune modification) [1,2]. Previously, studies have shown that normal skin cells can be reprogrammed to induce neuronal cells (iNCs) which could have important implications for the study of neural development, neurological disease modeling and regenerative medicine because isolation of neural stem cells from their niches within the subventricular and subgranular zones deep within the brain makes these stem cells a challenging source [3,4]. Combination of chemical substances and forced expression of transcription factors are important strategies. In the study of Wan et al., a cocktail of small molecules (valproic acid, CHIR99021, DMH1, Repsox, forskolin, Y-27632 and SP600125) converted human lung fibroblasts into functional neurons without the exogenous genetic factors after seven days of induction [5]. Later on, a combination of 16 small molecules (CHIR99021, LDN193189, SB431542, RG108, dorsomorphin, DMH1, parnate, SU5402, forskolin, Y27632, DAPT, purmorphamine, ISX9, IBET151, SU16F, and P7C3-A20) was used to induce human newborn foreskin fibroblasts into a neuronal morphology with positive TUJ1 immunostaining at 14 days of culture [6]. The viruses consisting of Ptf1a gene

were used to reprogram human induced neural stem cells (hiNSCs) and these cells could differentiate into neuronal and glial cell lineages in the hippocampus of adult mice after 1.5 months' transplantation [7]. Combination of transcription factors, i.e., Ascl1, Brn2, and Myt1l, was sufficient to convert mouse embryonic and postnatal fibroblasts into functional neurons in vitro. These induced neuronal (iN) cells expressed multiple neuron-specific proteins, generated action potentials, and formed functional synapses [8–10]. However, these methods inserted ectopic factors that might be more than enough for their global functions. Scientists are still seeking natural compromised methods which can maintain homeostasis of cellular function and biomimetic hydrogels could have been a method of choice because the Young's moduli of hydrogels could be adjusted to match the mechanical properties of brain tissue.

## 2. Why Fibroblasts

Fibroblasts are widely distributed and they are commonly utilized because they can be easily extracted from a patient using a safe and non-invasive skin biopsy and are easy to culture in laboratories; thus, it is of great benefit that human induced pluripotent stem cells (iPSCs) can be generated from fibroblasts. This iPSCs can allow easier generation of cells in the nervous system (neuron, astrocyte, oligodendrocyte, and microglia) called indirect reprogramming. The study from Zhang et al. proved that soft layered 3D culture system mimics the brain environment and accelerated maturation of neurons from human iPSC-derived NPCs, yielding electrophysiologically active neurons within three weeks. This work could draw the attention to the generation of iPSC from fibroblasts which might be one of the important steps for indirect conversion of fibroblasts to neurons using hydrogels [11].

Human iPSCs are similar to human embryonic stem (ESCs) cells in terms of morphology, proliferation, surface antigens, gene expression, epigenetic status, pluripotency, and telomerase activity [12,13]. Although many ES cell pluripotency-associated genes are co-regulated by Sox2 and Oct3/4, Sox2 may also cooperate with other transcription factors such as Nanog, c-Myc, Klf4 to activate transcription of pluripotency markers [14].

A previous study of Takahashi et al. demonstrated that iPSCs could be generated from adult human fibroblasts with four factors: Oct3/4, Sox2, Klf4, and c-Myc [15]. Genetically engineered fibroblasts have been successfully used to produce therapeutic proteins in animals, but sustained production of the proteins has not been achieved [16]. Cellular differentiation and lineage commitment are considered to be robust and irreversible processes during development. However, previous studies demonstrated that in vitro treatment of differentiated somatic cells with cell extracts induced the expression of pluripotent marker genes in a small population of cells [17–20], providing proof of concept, and supporting the hypothesis that differentiated cells may be reprogrammed. Differentiated somatic cells could be reprogrammed in ESC extract in vitro, which provided a new approach to decreasing differentiation levels in somatic cells without disturbing the DNA sequences [21]. Transient uptake of regulatory components from nuclear and cytoplasmic extract derived from ES cells by the nucleus of a reversibly permeabilized NIH/3T3 using streptolysin O could induce expression of Nanog, c-Myc, Klf4, Oct4, and Sox2 [21]. Further studies suggested that the ESC extract induced changes in DNA methylation. NIH/3T3 fibroblasts were treated with embryonic carcinoma cellular extract [21,22] and it was found that the expression of embryonic markers such as Sox2, klf4, c-Myc, and Oct4 were significantly increased after the treatment [21,22].

Direct isolation and generation of neurons in vitro from skin fibroblasts was discussed by Lindvall et al. [23]. Recent work from Vierbuchen et al. has shown that mouse and human fibroblasts could be reprogrammed to a pluripotent state with a combination of four transcription factors. Combinatorial expression of neural-lineage-specific transcription factors could directly convert fibroblasts into neurons, starting from a pool of 19 candidate genes. Combinations of only three factors, Ascl1, Brn2, and Myt1l, were sufficient to convert mouse embryonic and postnatal fibroblasts into functional neurons in vitro [8].

These induced neuronal (iN) cells expressed multiple neuron-specific proteins, generated action potentials and formed functional synapses [8]. Later, the study of Wang et al. indicated that NIH/3T3 fibroblasts could generate neurosphere-like, neuron-like, and even photoreceptor-like cells without any epigenetic modification or transcription factor requirements [24]. Hao et al. demonstrated that co-infection of Ascl1 with Pax6 directly reprogrammed fibroblast-like cells from adult human retinal tissues to the functional neuronal cells with mature morphology and active neural membrane electrophysiological activities [25,26].

Neurosphere-like cells were generated by floating cultures of NIH/3T3 fibroblasts in neural stem cell medium. These spheres expressed the neural progenitor markers nestin, Sox2, Pax6, and Musashi-1. When cultured in differentiating medium, cells from neurosphere expressed the neuronal markers beta-III tubulin and neurofilament 200 and the astrocytic marker, glial fibrillary acidic protein (GFAP). After treating the spheres with trans retinoic acid and taurine, expression of photoreceptor markers rhodopsin and recoverin were observed. It suggested that the differentiated non-neuronal cells NIH/3T3 fibroblasts may have the potential to be transdifferentiated into neuronal cells without recourse to epigenetic modifiers [24]. However, some groups propose that control and maintenance of gene expression is not only dependent on regulatory circuits of transcription factors, but is also dependent on epigenetic control. Thus, the modulation of epigenetic components has been exploited to reprogram fibroblasts into iPSCs [12,15].

One of these studies indicated that inhibition of DNA methylation and histone deacetylation could modify the epigenetic state of somatic cells. Neural-like cells were induced from reprogrammed fibroblasts and embryonic markers Sox2, klf4, c-Myc, and Oct4 were expressed in the reprogrammed NIH/3T3 fibroblasts. Moreover, exposure of the reprogrammed cells to all trans-retinoic acid (RA) medium elicited the generation of neuronal class III β tubulin- positive, neuron-specific enolase-positive, nestin-positive, and neurofilament light chain-positive neural-like cells [27]. Neuronal induction by retinoic acid was commonly used in ESC differentiation protocols. Two studies showed that this approach induced a population of neurogenic precursors [27,28]. Upon differentiation, RA-treated cells gave rise to a defined and developmentally restricted neuronal lineage. This role of RA in cell fate specification provided new perspectives for studying the radial glia-neuron transition and for generating homogenous populations of neurons from ES cells [28]. Histone deacetylation, Lys9 methylation, and hypophosphorylation of RNA polymerase II C-terminal domain were detected on this promoter after RA treatment which involved nucleosome recruitment and chromatin condensation [29]. Demethylation of specific CpG sites at the enhancer region could favor the displacement of MeCP2 from the heavily methylated rearranged during transfection (RET) enhancer region providing a novel potential mechanism for transcriptional regulation of methylated RA-regulated loci [30]. Moreover, fibroblasts showed great ability to survive after conversion to neurons. Avaliani et al. demonstrated that human lung fibroblasts directly converted to human neurons and they would survive for six months after transplanting into the adult rat hippocampus [31]. This evidence suggested that fibroblasts could be a treasure for nervous tissue regeneration in the recent era; however, it is still essential to evaluate the safety of potential methods. Some important results from our previous studies are the specific characteristics of each cell type. For example, the use of human gingival cells is convenient because this cell type is easy to harvest and provides a massive fibroblast population and, perhaps, stem cell population. This cell type showed the possibility of transdifferentiation to neuron [32], however, the potential of transdifferentiation to neuron is varied depending on the individual donor.

## 3. How Does Epigenetics Modulate Cellular Reprogramming and Lineage Choice?

Epigenetics is the heritable modulation of gene activity that does not alter the underlying DNA sequence ("epigenetics" means "besides genetics" and is used to describe heritable alterations of phenotypic traits that are not based on changes in the DNA sequence [33]). Cell identity is defined by the interplay of DNA methylation, histone modifications, mi-

croRNAs, and DNA-binding proteins, which shapes the transcriptional profile of each cell type by modulating the chromatin landscape [33]. Liu et al. demonstrated that combining global manipulation of DNA methylation and histone acetylation together with the expression of oligodendrocyte-specific transcription factors were not sufficient to switch the identity of fibroblasts into myelin gene-expressing cells [34]. This means reprogramming of fibroblasts into myelin gene-expressing cells requires more than transcriptional activation, but also needs chromatin manipulations to go beyond histone acetylation and DNA methylation [35–37]. This statement was supported by Singhal et al. when a mechanistic insight into BAF complex function was studied. It revealed that BAF complex components achieved a euchromatic chromatin state and enhanced binding of reprogramming factors onto key pluripotency gene promoters, thereby enhancing reprogramming [35].

The importance of epigenetic gene control has now been widely studied for various biological processes, including cell differentiation, stem cell plasticity, cell cycle control, dosage compensation, and stabilization of genome integrity. Epigenetic gene regulation involves the alteration of chromatin structure through histone modifications, i.e., methylation, acetylation exchange with histone variants, and DNA methylation. In addition to histone modifications and DNA methylation, non-histone chromosomal proteins such as ATP-dependent chromatin remodelers are also essential controllers of chromatin structure and function. These chromatin modifications and the interactions of non-histone chromosomal proteins are mitotically heritable, but still retaining the reversible characteristics allowing context-dependent changes to gene expression [38].

Transdifferentiation is the next step to be considered. Two transdifferentiation models have been proposed by Jopling et al. The first model proposes that a pluripotent transition phase is required. To transdifferentiate, a cell must first undergo dedifferentiation to a precursor stage before it can enter the new lineage and subsequently differentiate. In the second model, cells directly transdifferentiate to form the new cells, in some cases passing through an unnatural intermediate phase in which two genetic programs are active at the same time [39]. The 'unnatural' intermediate cells become primed which leads to a relaxation of previously restricted chromatin, allowing the exogenous transcription factors access and beginning the induction of pluripotency. Additionally, Nanog is also required to allow the cells to reach the pluripotent state; without Nanog, pluripotency would not happen [40]. Once the cell has reached this point, the endogenous program comes into play to maintain pluripotency and self-renewal [39].

The minimum number of factors required for iPS cell generation in maintenance of pluripotency was reviewed by Lewitzky et al. [14]. Oct3/4 is a tightly regulated transcription factor that is associated with a large number of target genes implicated in maintenance of pluripotency.

In this review we discuss if it is possible to directly induce fibroblasts to neurons using intrinsic properties of culture surfaces. Surfaces modulate cell behavior and thus have the potential to act as non-invasive tools. McMurray et al. demonstrated the potential of the nanostructured surface to retain mesenchymal stem-cell phenotype for prolonged times (it is noted that adult stem cells rapidly and spontaneously differentiate in tissue culture flasks). Furthermore, the study implicated a role for small RNAs in repressing key cell signaling and metabolomic pathways [41]. The combination of both transcriptomic and epigenomic profiling offers insight into different levels of gene regulation, transcription factor binding motifs, DNA and chromatin modifications, and how each component is coupled to a functional output. Chromatin remodelers and transcription factors are in close communication via recognition of post-translational histone modifications [42]. Successful reprogramming of differentiated human somatic cells into a pluripotent state would allow the creation of specific cell lineage to use as a tool for biomedical sciences and therapeutic medicine.

## 4. Biomimetic Biomaterial for Fibroblast Reprogramming

Previously, biomimetic surfaces presenting the active peptide domains of various extracellular matrix (ECM) proteins can be used to regulate neural differentiation in vitro [43]. In this context, hydrogels can be a promising candidate for transdifferentiation of fibroblasts to neurons. The special characteristics that make hydrogels become biomaterials of choice for these purposes are modifiable chemical properties, biocompatibility, elasticity, the capability to act as a growth medium, and the ability to mimic the ECM [44]. Moreover, encapsulation of fibroblasts in a hydrogel substrate before polymerization can be performed and this facilitates the arbitrary design of the shapes of 3D constructs [45].

A study from Pelham et al. demonstrated that, on ligand-coated gels of varied stiffness, epithelial cells and fibroblasts were reported to detect and respond distinctively to soft versus stiff substrates [46].

Photocurable hydrogels are of interest because of their potential use in regenerative medicine due to their ability to incorporate specific growth factors and their ability to manipulate fibroblast behavior [47]. Our previous results present that Human mesenchymal stem cell (hMSC) culture on phototunable hydrogels developed a highly arcuate, branched morphology (Figure 1) within 17 days.

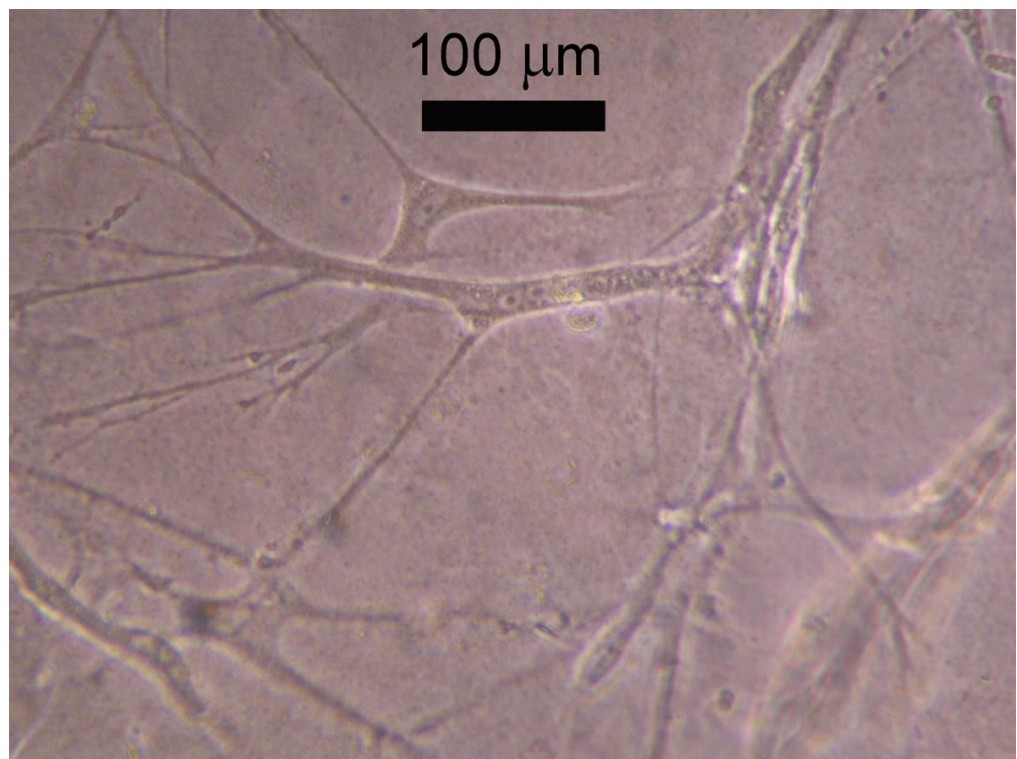

**Figure 1.** Human mesenchymal stem cells (hMSCs) on ultra-soft hydrogel: hMSCs could be cultured on ultra-soft hydrogel for 17 days and the extended morphology of healthy cells was observed.

Biomechanical factors such as extracellular matrix elasticity are known to influence cell functions. Studies by Kantawong et al. revealed that a soft environment promoted changes in the expressions of neuronal genes (TUBB3 and NSE) and redox genes (TRX1, SOD1, SOD2, PRX2, GSTT1, and GSTP1) in human adipose-derived stem cells (ADSCs). It was found that the TUBB3 gene was significantly upregulated on soft hydrogel compared to the control condition for tissue culture polystyrene, indicating that the neuronal gene expression of ADSCs could be achieved on soft hydrogel without the addition of any supplement. The lesson from this study is the problem about the reproducibility of photocurable hydrogels because many sensitive factors need to be precisely controlled in the fabrication of hydrogels such as UV-light intensity, the preparation of styrenated

gelatin and the Atomic Force Microscope (AFM) technique using for Young's modulus determination [48].

Our preliminary data, in agreement with the literature, suggests that hydrogels could become a biomaterial of choice to use as substrates, scaffolds, and/or encapsulants for stem cells due to their tissue-like and tunable material properties. Photo cross-links designed within their network enable gels to be engineered with desirable mechanical and biodegradable properties [49]. Agarose-based 3D hydrogels containing stem cell differentiation factors for promoting lineage commitment of retinal precursor cells was proposed by Wylie et al. [50]. The development of this field of research will lead to the creation of synthetic ECM surfaces tailored to facilitate generation of iPS/transdifferentiated cells and will be of direct relevance to tissue engineering, regenerative medicine, and drug design.

Glutaraldehyde (GA) is a well-known substance because it is available and inexpensive, and the elasticity of hydrogels can be modified by rendering the percentage of added GA. The study by Distantina et al. used GA as the crosslinking agent in the preparation of Kappa carrageenan film. It was found that the swelling degree of Kappa carrageenan film in water decreased with increasing GA concentration [51]. However, the experiences from a previous study by our group indicated that improper mixing technique can cause non-homogenous hydrogel matrix and the leftover GA residuals can interfere the freeze-drying process by generation of cracks on dried hydrogel scaffolds. Since the shelf-life of fabricated hydrogels is very important, however, the freeze-drying process cannot be avoided. Moreover, GA residuals can be harmful to the cultured cells. This problem was solved by soaking hydrogels in water before lyophilization to remove all GA residuals [52,53]. Various methods and principles about hydrogel crosslinking were provided by Hu et al. [54].

Currently, 3D imaging technology has been adopted in the field of neural tissue engineering. 3D printing can fabricate scaffolds based on the imaging data of the patient's defect with mimicking the microstructure of natural tissue in morphology. A review on 3D bioprinting for neural tissue engineering was provided by Yu et al. [55]. 3D printing technology might provide great advantages in the construction of peripheral neural scaffolds, however, the other technologies such as crosslinking and controlling elasticity still need to be incorporated to achieve the suitable scaffold for each situation.

It is known that substrate rigidity influences contractility, motility, and cell spreading [56]. Interplay between physical and biochemical signals results in the contractility and cell signaling as cells exert less tension on softer surface. Discher et al. demonstrated that cells crawled faster on collagen-coated gels, causing an accumulation of cells toward the stiff end of a soft-to-stiff gradient gel [56]. Nonlinear responses to surface rigidity were observed from adhesions, cytoskeleton organization, tractions exerted on the substrate, and other cellular processes [56,57]. Specific cell surface receptors mediate these interactions. The largest family of receptors, which mediates cell adhesion to fibronectins, laminins, and collagens, is called the integrins. Several other cellular receptors have also been involved in binding to various matrix components [58].

Banerjee et al. described that the rate of proliferation of neural stem cells decreased with increase in the modulus of the hydrogels. Moreover, expression of the neuronal marker β-III tubulin was detected within the softest hydrogels, which possessed an elastic modulus comparable to that of brain tissue [59]. A study by Kantawong et al. showed that addition of nano-hydroxyapatite (HA) could enhance cell adhesion on a soft substrate and, maybe together with the releasing of $Ca^{2+}$, induced a cell response to change gene expression and differentiation. Gelatin with 1.0 mg/mL supported neuronal gene expression in HMSCs, compared with pure gelatin and gelatin with 0.5 mg/mL of HA [53]. These studies supported that the influence of modulus on NSC differentiation that control stem cell fate would be possible for applications in reprogramming and cellular transdifferentiation.

Electrical signals are the basis of information transfer in the nervous system, thus, hydrogels that have electrical conductivity might be one stratregy for fibroblast transdifferentiation. Alginate is a polysaccharide extracted from brown algae which is biocompatible,

non-toxic, and non-immunogenic [60]. Alginate is widely used in various types of tissue engineering [45,61,62]. Alginate-based hydrogels present electrical conductivity and encapsulated fibroblasts dominate the conductance, as shown by the increasing of total conductivity of cell–hydrogel constructs [60]. Encapsulated neural stem cells in alginate hydrogel were exposed to electrical stimuli using oscillatory fields. It was found that neuronal differentiation was either enhanced or suppressed depending on the electric field frequency and culture time [63].

Elasticity is the important property of hydrogels which make hydrogels able to work as shock absorbers and hydrogels can transfer stress-strain to cells. Human Schwann cells were encapsulated in fish gelatin methacrylamide hydrogels and tensile forces were applied during cell culture. It was shown that tensile forces applied to hydrogels enhanced proliferation and differentiation through PI3K/AKT pathway compared to static cultures [64]. Moreover, the elasticity of hydrogel can be manipulated by the combination between natural polymer and synthetic polymer. A combination of 7% gelatin, 0.5% PVA, and 0.1% chitosan improved the Young's modulus, pore size, swelling rate, and degradation rate of the scaffolds for cultivation of NIH/3T3 by upregulation of type IV collagen compared to plain gelatin scaffolds [52].

Polyvinyl alcohol (PVA) is one of the most popular synthetic polymers for scaffold fabrication. Nevertheless, the elasticity of the standalone PVA hydrogel could not match the elasticity of soft tissue since PVA always acts as a stiff membrane and has limited hydrophilicity [65]. The study of Muduli et al. indicated that human embryonic stem cells did not adhere well to soft PVA hydrogels immobilized with oligo-vitronectin, whereas they adhered well to PVA hydrogel dishes with elasticities greater than 15 kPa. These results indicated that biomaterials such as PVA hydrogels should be modified to possess optimal elasticity to facilitate cell attachment [66]. Normally, a suitable Young's modulus of biomaterials for neuronal differentiation should be <1 kPa [67]. Thus, it is necessary to incorporate other substances with PVA for modulation of the elasticity. This knowledge could be applied when either natural or synthetic polymers are employed. For example, the mixture of chitosan–alginate hydrogel could enhance olfactory ensheathing cells and neural stem cell proliferation [68].

In the preliminary study by Kantawong et al., polyacrylamide gels were prepared from 40% acrylamide stock solution, 2% bis-acrylamide stock solution, and the polymerization was induced by ammonium persulfate (modified from the method of Kippert et al.) [69]. Polyacrylamide gels were coated with 4% BSA using 0.5 mg/mL Sulfo-SANPAH as the crosslinker and the Young's modulus of the prepared polyacrylamide gels was <1 kPa. Polyacrylamide gels were employed for the cultivation of human adipose-derived stem cells (ADSCs) for one week with positive TUBB3 staining, as shown in Figure 2. However, polyacrylamide gel prepared by this method is not a biodegradable and biocompatible material, so this study was not continued.

Integration of chemical and biological substances into hydrogels can make hydrogels become better bioactive materials for neuronal differentiation. Hydrogels loaded with the nerve growth factor enhanced the proliferation and differentiation of neuronal stem cells [70]. A recent study by Kantawong et al. found that *Gynura divaricata* crude extract can enhance neuronal differentiation of human gingival cells [32]. Thus, our team aims to combine this herbal extract with hydrogels for fibroblast transdifferentiation to neuron in the future study.

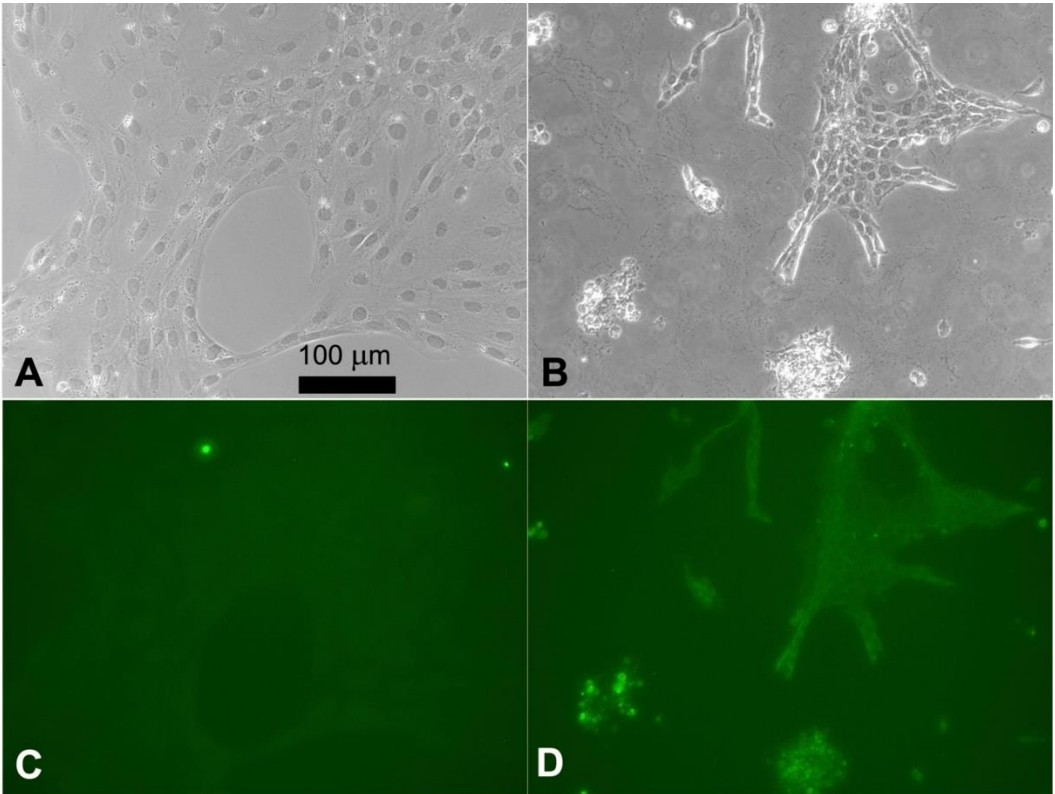

**Figure 2.** Human adipose-derived stem cells (ADSCs) on polyacrylamide gels. (**A**,**C**): ADSCs cultured on tissue culture plate for one week showed negative TUBB3 staining. (**B**,**D**): ADSCs cultured on polyacrylamide gel for one week showed positive TUBB3 staining.

## 5. Mechanotransduction and Epigenetics

Understanding the physical effects of the in vivo microenvironment has significant implications for synthetic biomimetic surfaces designed for potential use in generation of iPSCs. Forces from local matrix stiffness have effects on the cells' ability to spread which has important implications for development, differentiation, disease, and regeneration. Cells commit to the lineage specified by matrix elasticity. The previous study by Kantawong et al. indicated that ADSCs cultured on photocurable hydrogel for 21 days adopted a neural-like morphology as shown in Figure 3. Adhesion complexes and the cytoskeleton play key roles in molecular pathways, and which contractile forces are transmitted through transcellular structures. Inhibition of non-muscle myosin II blocked all elasticity-directed lineage without strongly disturbing other aspects of cell function and shape [57]. Focal adhesions are key structures that interact with the material surface and are central to many signaling cascades, i.e., G-protein, Rho, Rac, and Cdc42, which modulate cell 'sensing', shape, and contractility, as well as mitogen-activated protein kinases (MAPKs) such as extracellular-signal-regulated kinase (ERK) and p38 MAPK. These signaling cascades have downstream effectors such as a wide range of transcription factors which are involved in cell survival, proliferation, and differentiation.

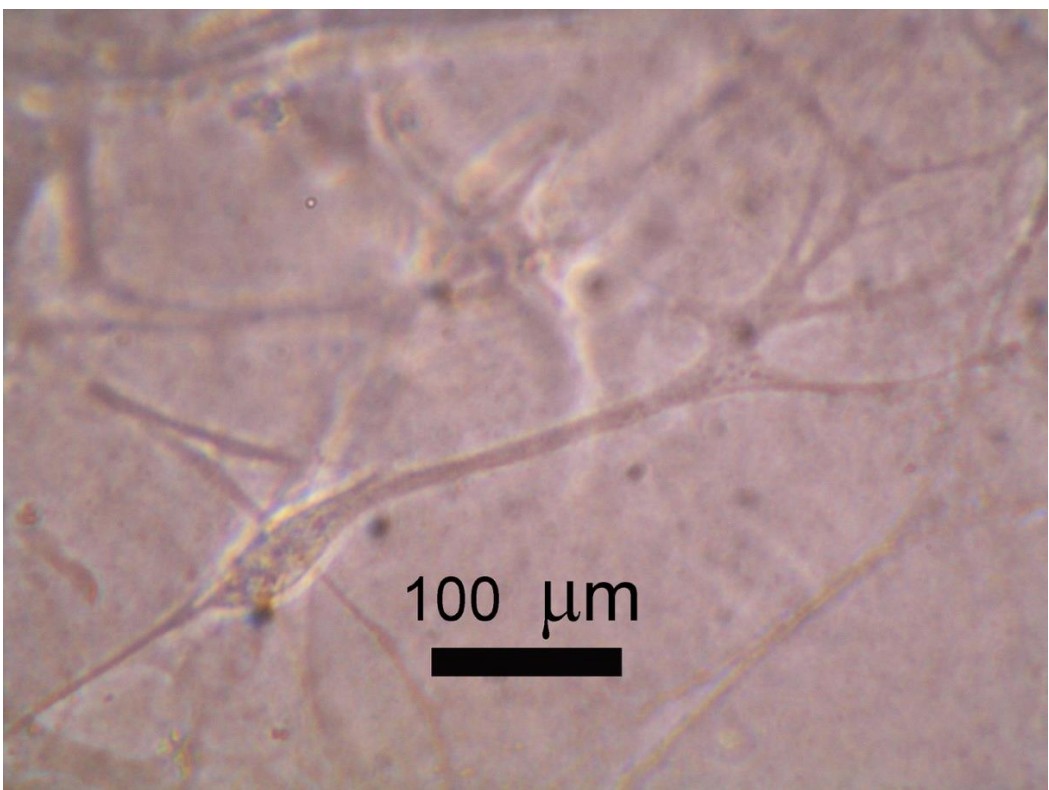

**Figure 3.** ADSCs on ultra-soft hydrogel: ADSCs cultured on photocurable hydrogel for 21 days adopted a neural-like morphology induced by matrix elasticity which happens via material-mediated mechanotransduction.

Another way of modulating transcription factor activity is through direct cytoskeletal signaling [71]. Cytoskeletal signaling happens when focal adhesion-mediated cytoskeletal re-arrangements lead to distortion of the nucleus speculatively altering heterochromatin/euchromatin balance, hence physically opening up or closing down gene availability to transcription factors and polymerase enzymes. In this context, the transcription factors are key to phenotypical control through material-mediated modulation of focal adhesions [72–74]. These considerations lead to the key question: Is it possible to use biomimetic material to generate iPSCs or to permit transdifferentiation from differentiated cells? The answer to these questions is that we have to consider previous studies (which employed epigenetic modifications and exogenous transcription factors) of what happens in material-mediated mechanotransduction.

Chemical, topographical, and stiffness (Young's modulus) modifications can influence adhesion size, shape, and number and thus change cytoskeletal arrangements. The nucleus is connected to the cytoskeleton via the lamin nucleoskeleton and the lamins are closely associated with the telomeres of interphase chromosomes [75–77]. The extracellular environment regulated mechanical changes being transmitted through the cytoskeleton to the nucleus, as eloquently described by Wang et al. [78]. Passing these mechanical changes to the nucleus may change the location of key genes to transcription factors, thus altering genomic expression patterns and cell phenotype [79,80]. This phenomenon suggests a fundamental role for mechanosensing in mammalian development and illustrates that the mechanical environment should be taken into consideration for use in engineering implantable scaffolds and in producing iPSCs for experimental modeling purposes such as drug discovery, and even, potentially, for producing therapeutically relevant cells in vitro [56,57,81].

Fluid flow can generate shear stresses which is considered one of the important mechanical stimuli for neuronal differentiation [82] Fluid flow can induce biomimetic mechanotransduction because, in the human body, cerebrospinal fluid flow increases cell

proliferation and Erk activation of neural stem cells. Cerebrospinal fluid flow is also involved in polarizing the ependymal cells and directing neuroblast migration [83]. A bioreactor system combined hydrodynamic and electrical stimuli on cells in 3D scaffolds showed better improvements in neural differentiation than individual hydrodynamic or electrical stimulus. This result indicated that multidisciplinary of mechanical stimuli can be applied together. The mechanical stimuli play important roles in epigenetic modification, as presents in Figure 4.

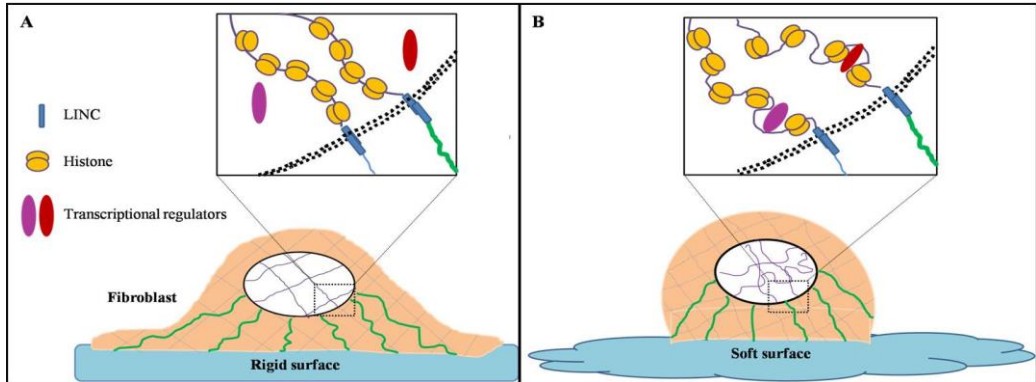

**Figure 4.** Schematic model of surface mechanotransduction modulating chromatin structure: Biomimetic surfaces can modulate chromatin structure in the same way as extracellular matrices can (in a natural way). (**A**) Fibroblasts on rigid surface: Fibroblasts on a rigid surface may have less flexible chromatin structures which does not allow transcriptional regulators to access the DNA. (**B**) Fibroblasts on soft matrix: Fibroblasts on soft matrix (e.g., extracellular matrix (ECM) of the brain) have a more flexible chromatin structure allowing specific transcriptional regulators to work and desirable gene expressions (for neuronal regeneration) are achieved.

It is thus possible to start considering epigenetics and mechanotransduction acting together if we consider the nucleus itself acting as a mechanosensory, as deformations (from force transmitted through the cytoskeleton to the nucleoskeleton) can influence chromatin conformation. Applied force is transmitted to the DNA through the cytoskeleton by nuclear lamins and nuclear envelope receptor complexes [78,84]. Such a connection might then directly modulate gene expression by inducing conformational changes in chromatin either by altering the nature of the protein complexes at the telomeres of chromosomes, or by changing the activity of DNA-remodeling enzymes [85]. The resultant modifications in global patterns of chromatin, histone acetylation, and transcriptional regulation of gene expression would regulate phenotype. Thus, a connection between components of the linker of nucleoskeleton and cytoskeleton (*LINC*) *complex* and changes in chromatin structure in response to mechanical cues could dynamically alter gene expression in response to exogenous force [86,87].

Chromatin conformations may also determine the cells' regenerative ability. The chromatin structure reflects the transcriptional state. Flexible chromatin has both active and suppressive histone modifications (bivalent domain) and/or low level of DNA methylation, which leads to the dormant state of target genes [38]. In contrast, the inactive chromatin would be dominated by the suppressive histone marks and/or hypermethylated DNA, resulting in more closed chromatin. The switch of transcription state from inactive to active will be more feasible at flexible chromatin than at inactive chromatin [38].

Differentiated somatic cells can be reprogrammed to generate induced pluripotent stem (iPS) cells via overexpression of a cocktail of transcription factors such as Oct3, Sox2, Klf4, and c-Myc or Oct3, Sox2, Nanog, and Lin28 [88]. However, overexpression of transcription factors is worthless if chromatin structure does not support binding of those transcription factors. We postulate that manipulation of chromatin structure by mechanotransduction may work in a similar way to chromatin remodeling drugs [86]. A study by Kantawong et al. demonstrates that the NIH/3T3 cell culture on the gelatin with hydroxya-

patite and pig brain extract express a set of transcription factors (i.e., NFIa, NFIb, Ptbp1, and SOX9), indicating that the cells are differentiated into an astrocytic lineage. The cells might be passed through the neural stem cell (neural progenitor) state because the NIH/3T3 cells expressed the Klf4, a pluripotency marker which is significantly upregulated on the gelatin with hydroxyapatite and pig brain extract during the first week of the cell culture. The PI3K/Akt activation is involved in the early steps of the reprogramming process as it is increased in NIH/3T3-cell-cultured gelatin with hydroxyapatite and pig brain extract [89]. This evidence strongly supports that hydrogels can modulate transcription factors to cause reprogramming. We could hypothesize that biomimetic surfaces might be less invasive than the drugs and modulate chromatin in a more targeted fashion. Biomimetic surfaces might remodel chromatin structure in a similar mechanism to that of the ECM, hence allowing other transcription factors to function as natural reprogramming and natural transdifferentiation (Figure 4).

## 6. Summary

Fibroblasts can be used for tissue regeneration in the future. By manipulation of epigenetic and transcription factor systems, fibroblasts present the possibility to be reprogrammed for the generation of iPSCs and transdifferentiation into neurons. Biomimetic surfaces such as hydrogels could potentially be a valuable tool for generation of iPSCs and transdifferentiation of somatic cells. Interaction forces between cells and surfaces pass through the nucleus which works as a mechanosensor. Tensile force and shear stress can be applied to hydrogels which act as the transporter of these stimuli to cells. Nucleus deformations can influence chromatin conformation that might directly modulate gene expression by altering the nature of the protein complexes resulting in modified global patterns of chromatin histone acetylation and transcriptional regulation of gene expression. The major concern about reprogramming is the adverse effects. Using epigenetic modifiers can put stresses on cells during reprogramming which can lead to the mutation of stress regulatory genes which are prone to tumor formation. Instead of trying to force a cell with epigenetic modifiers, transfected transcription factors, or chromatin-remodeling drugs, it may be safe to try more natural tools, like biomimetic biomaterials which are similar to what occurs during natural reprogramming and transdifferentiation [90]. If it is necessary, further manipulation such as increased expression of particular transcription factors could be used as a parallel technique to help permit functional transdifferentiation. Incorporation of chemical or biological substances can improve the properties of hydrogels to become more bioactive. Combination of various polymer types can modulate the desirable elasticity. We end by stating that we recognize that materials-directed transdifferentiation is a moot point, despite having been described in major papers [57,91]. We run with these major observations and focus on the potential of fibroblasts and hydrogels to help neural regenerative strategies.

**Funding:** This research received no external funding.

**Institutional Review Board Statement:** Not applicable.

**Informed Consent Statement:** Not applicable.

**Data Availability Statement:** The data that support the findings of this study are available from the corresponding author on request.

**Acknowledgments:** The author gratefully extends her thanks to Matthew Dalby, Institute of Molecular Cell & Systems Biology, Glasgow University and Satoru Kidoaki, Department of Applied Molecular Chemistry, Kyushu University for their comments and suggestions. The author would like to give a special thanks to Thasaneeya Kuboki for all of her helps in photocurable hydrogels fabrication.

**Conflicts of Interest:** The author declares no conflict of interest.

## Abbreviations

| | |
|---|---|
| BAF complex | BRM-associated factors |
| $Ca^{2+}$ | Calcium ion |
| DNA | Deoxyribonucleic acid |
| HA | Hydroxyapatite |
| hiNSCs | Human induced neural stem cells |
| hMSCs | Human mesenchymal stem cells |
| iPSCs | Induced pluripotent stem cells |
| NPCs | Neural progenitor cells |
| NSC | Neural stem cells |
| NSE | Neuron-specific enolase |
| Ptf1a | Pancreas Associated Transcription Factor 1a |
| RET | Rearranged during transfection |
| RNA | Ribonucleic acid |
| TUBB3 | Tubulin Beta 3 Class III |

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
