# Peer review of "The Potential of Fibroblast Transdifferentiation to Neuron Using Hydrogels"

_processes, doi:10.3390/pr9040632_

Round 1

Reviewer 1 Report

This review is focusing mainly on the fibroblasts properties, microenvironment and behavior. Only a small part dealing with hydrogels and their potential use in regenerative medicine.

  1. Would suggest to modify the title to reflect more exactly the content of the review. Another way is to complete the review with more studies included fibroblasts suppoted by hydrogels. May be including a systematic characterization focused on the hydrogel type (PVA, Chitosan, Ca alginate and so on) would increase to the value of this review.
  2. A better highlight why the hydrogels are suitable, which is the characteristics who made them proper candidates, should be also introduced.
  3. May be a chapter about other the synthetic materials, which are used for fibroblast supporting is also welcomed.
  4. The introduction and summary parts need to be updated in agreement with the newly introduced parts.

Author Response

Response to reviewer 1

Thank you for your comments. I tried to follow the suggestions as shown by yellow highlighting the texts. However, I realized that this review is still not good enough. The reviewer can comment again and I will try to revise it.

  1. Would suggest to modify the title to reflect more exactly the content of the review.

The title was modified to;

The Potential of Fibroblasts Transdifferentiation to Neurons Using Hydrogels

  1. Another way is to complete the review with more studies included fibroblasts supported by hydrogels. May be including a systematic characterization focused on the hydrogel type (PVA, Chitosan, Ca alginate and so on) would increase to the value of this review.

Lines 299-337 was added;

Electrical signals are the basis of information transfer in the nervous system, thus, hydrogels that have electrical conductivity might be one stratregy for fibroblast transdifferentiation. Alginate is a polysaccharide extracted from brown algae which is biocompatible, non-toxic and non-immunogenic [60]. Alginate is widely used in various types of tissue engineering [45,61,62]. Alginate-based hydrogels presented electrical conductivity and encapsulated fibroblasts dominated the conductance as shown by the increasing of total conductivity of cell-hydrogel constructs [60]. Encapsulated neural stem cells in alginate hydrogel were exposed to electrical stimuli using oscillatory fields. It was found that neuronal differentiation was either enhanced or suppressed depending on the electric field frequency and culture time [63].  

Elasticity is the important property of hydrogels which make hydrogels can work as shock absorbers and hydrogels can transfer stress-strain to cells. Human Schwann cells were encapsulated in fish gelatin methacrylamide hydrogels and tensile forces were applied during cell culture. It was shown that tensile forces applied to hydrogels enhanced proliferation and differentiation through PI3K/AKT pathway compared to static cultures [64]. Moreover, elasticity of hydrogel can be manipulated by the combination between natural polymer and synthetic polymer. Combination of 7% gelatin, 0.5 % PVA and 0.1% chitosan improved Young’s modulus, pore size, swelling rate and degradation rate of the scaffolds for cultivation of NIH/3T3 by upregulation of type IV collagen compared to plain gelatin scaffolds [52].    

Polyvinyl alcohol (PVA) is one of the most popular synthetic polymers for scaffold fabrication. Nevertheless, the elasticity of the standalone PVA hydrogel could not match the elasticity of soft tissue since PVA always acts as a stiff membrane and had limited hydrophilicity [65]. The study of Muduli et al. indicated that human embryonic stem cells did not adhere well to soft PVA hydrogels immobilized with oligo-vitronectin, whereas they adhered well to PVA hydrogel dishes with elasticities greater than 15 kPa. These results indicated that biomaterials such as PVA hydrogels should be modified to possess optimal elasticity to facilitate cell attachment [66]. Normally, suitable Young's modulus of biomaterials for neuronal differentiation should be < 1 kPa [67]. Thus, it is necessary to incorporate other substances with PVA for modulation of the elasticity. This knowledge could be applied when either natural or synthetic polymers will be employed. For example, the mixture of chitosan-alginate hydrogel could enhance olfactory ensheathing cells and neural stem cells proliferation [68].

Integration of chemical and biological substances into hydrogels can make hydrogels become better bioactive materials for neuronal differentiation. Hydrogels loaded with the nerve growth factor enhanced the proliferation and differentiation of neuronal stem cells [69]. Recently study by Kantawong et al. found that Gynura divaricata crude extract can enhance neuronal differentiation of human gingival cells [32]. Thus, our team aims to combine this herbal extract with hydrogels for fibroblast transdifferentiation to neuron in the future study.    

  1. A better highlight why the hydrogels are suitable, which is the characteristics who made them proper candidates, should be also introduced.

Lines 59-64 were added to the introduction;

because the stiffness and the porosity of hydrogels can be controllable. The biodegradability and biocompatibility of the hydrogel are well acknowledged. Mechanical properties of hydrogels are suitable for induction of neuronal morphology. Hydrogels are easy to fabricate as drug delivery scaffold. Additionally, hydrogel can be slowly degraded while enhancing cells proliferation and cell differentiation with compatibility to native tissue.

  1. May be a chapter about other the synthetic materials, which are used for fibroblast supporting is also welcomed.

The sentences mentioned about synthetic materials were included in the revision.

  1. Summary part need to be updated in agreement with the newly introduced parts.

Summary part was re-written.

Summary

Biomimetic surfaces could potentially be a valuable tool for generation of iPSCs and transdifferentiation of somatic cells. Interaction forces between cells and surfaces pass through the nucleus which works as a mechanosensor. Tensile force and shear stress can be applied to hydrogels which act as the transferer of these stimuli to cells. Nucleus deformations can influence chromatin conformation that might directly modulate gene expression by altering the nature of the protein complexes resulting in modified global patterns of chromatin histone acetylation and transcriptional regulation of gene expression.  The major concern about reprogramming is the adverse effects. Using epigenetic modifiers can put stresses on cells during reprogramming which can lead to the mutation of stress regulatory genes which prone to tumor formation. Instead of trying to force a cell with epigenetic modifiers, transfected transcription factors or chromatin remodeling drugs, it may be safe to try more natural tools, like biomimetic biomaterials which are similar to what occurs during natural reprogramming and transdifferentiation [86]. If it’s necessary, further manipulation such as increased expression of particular transcription factors could be used as a parallel technique to help permit functional transdifferentiation. Incorporation of chemical or biological substances can improve the properties of hydrogels to become more bioactive. Combination of various polymer types can modulate the desirable elasticity. We end by stating that we recognize that materials directed transdifferentiation is a moot point, despite having been described in major papers [51,87]. We run with these major observations and focus on the potential of fibroblasts and hydrogels to help neural regenerative strategies.

Reviewer 2 Report

This review article (Manuscript number processes-1127749-peer-review-v1) presents information about the application of hydrogels as non-invasive tools to direct transdifferentiation.

On the whole, the manuscript is quite fairly well-written and but not logically arranged. The overall originality of the review concept used here is medium-high. The topic of this review is interesting, and the manuscript constitutes an interesting review concerning the development of new hydrogels. Therefore, I would recommend the publication of this paper in the Processes on the condition a fundamental major revision of the manuscript will be carried out and the following points will be taken into consideration.

Detailed comments:

  1. The abstract needs to be well written with future prospects of the work and describe in short the concept of hydrogels used in neural regenerative strategies.
  2. More detailed advantages of the present field must be mentioned in the Introduction. There should be a critical discussion of the state of knowledge of the field - what the previous studies establish definitively, what the tentative interpretations are, and what specific fundamental research questions still need to be studied. 
  3. Furthermore, the introduction should be worked out - so as to show the full state of knowledge on different polymerization methods used for preparation of hydrogels. The author needs to explain the controlled/living radical polymerization possibility. Extension to look at these issues and also provide other techniques should also be provided. 
  4. It would be better to tabulate the advantage of the system compared to the other initiating systems in a table at least for the controlled radical polymerization.
  5. The introduction appears to be a collection of data from research papers, however, the author's self-opinion is of importance while drafting a section of this type.
  6. The conclusion reflects an overall summary of the field with further extension and includes future prospective - I would suggest clarifying this section.
  7. The style and grammar leave much to be desired in many places, some parts of the text are difficult to understand.

After completing the above-mentioned corrections this work will be more readable. Therefore, it will be useful for the readers of the Processes.

Author Response

Response to reviewer 2

Thank you for your comments. I tried to follow the suggestions as shown by yellow highlighting the texts in the manuscript. However, I realized that this review is still not good enough. The reviewer can comment again and I will try to revise it.

Detailed comments:

  1. The abstract needs to be well written with future prospects of the work and describe in short, the concept of hydrogels used in neural regenerative strategies.

Abstract was re-written and the graphical abstract was added.

Abstract: Currently there is a big drive to generate neurons from differentiated cells which would be of great benefit for regenerative medicine, tissue engineering and drug screening. Most studies used transcription factors, epigenetic reprogramming and/or chromatin remodeling drugs which might reflect incomplete reprogramming or progressive deregulation of the new program. In this review, we present a potential different method for cellular reprogramming/transdifferentiation to potentially enhance regeneration of neurons. We focus on the use of biomaterials, specifically hydrogels, to act as non-invasive tools to direct transdifferentiation and we draw parallel with existing transcriptional and epigenetic methods. Hydrogels are attractive materials because the properties of hydrogels can be modified and variable of natural and synthetic substances can be employed. Incorporations of ECM substances and composite materials allow mechanical properties and degradation rate can be controlled. Moreover, hydrogels in combinations with other physical and mechanical stimuli such as electric current, shear stress and tensile force will be mentioned in this review.

  1. More detailed advantages of the present field must be mentioned in the Introduction. There should be a critical discussion of the state of knowledge of the field - what the previous studies establish definitively, what the tentative interpretations are, and what specific fundamental research questions still need to be studied. 

Advantages of the present field were added with more references.

  1. Furthermore, the introduction should be worked out - so as to show the full state of knowledge on different polymerization methods used for preparation of hydrogels. The author needs to explain the controlled/living radical polymerization possibility. Extension to look at these issues and also provide other techniques should also be provided. 
  2. It would be better to tabulate the advantage of the system compared to the other initiating systems in a table at least for the controlled radical polymerization.
  3. The introduction appears to be a collection of data from research papers, however, the author's self-opinion is of importance while drafting a section of this type.
  4. The conclusion reflects an overall summary of the field with further extension and includes future prospective - I would suggest clarifying this section.
  5. The style and grammar leave much to be desired in many places, some parts of the text are difficult to understand.
  • For issues 2-6, I tried to insert the details that the reviewer asked for but they may not cover every issue. If the reviewer needed more details, please suggest me.
  • About the gramma, I admit that it is very hard for me because it is not my native language. However, I tried to correct it but, if it is too bad, I will try again.

Reviewer 3 Report

The review is well organized and the theme of potential different method for cellular reprogramming/transdifferentiation to potentially enhance regeneration of neurons is well discussed. 

Author Response

Dear Reviewer

Thank you very much.

Round 2

Reviewer 1 Report

The present review was improved after the first round of corrections, but there are some issues to fix before the publication:

Line 17 – The first abbreviation of ECM should be made.

Lines 59-64 – this part should be rephrase. For me the newly introduced part are composed only by sentences without a logical connection.

Line 218 – “hydrogels can be the best candidate” –  biohydrogels can be excellent/good candidates – in my opinion the best is too much.

Lines 258 – 263 – I would delete the general sentence about crosslinking and I would rephrase the sentence starting from line 259 like:  Glutaraldehyde (GA) is a well-known crosslinker….

The summary is to specific, it is not covering all the chapters presented in this review. The hydrogels are mentioned only ones. Please complete the summary at least with one phrase about the findings from every chapter.

Please recheck the hole manuscript for typos (ex: line 259 substanc)

Author Response

Comments and Suggestions for Authors

Dear Reviewer

Thank you very much for your suggestion.

  1. Line 17 – The first abbreviation of ECM should be made.

extracellular matrix (ECM)

  1. Lines 59-64 – this part should be rephrased. For me the newly introduced part are composed only by sentences without a logical connection.

Scientists are still seeking for natural compromised methods which can maintain homeostasis of cellular function and biomimetic hydrogels could have been a method of choice because the Young’s moduli of hydrogels could be adjusted to match the mechanical properties of brain tissue.

  1. Line 218 – “hydrogels can be the best candidate” –  biohydrogels can be excellent/good candidates – in my opinion the best is too much.

hydrogels can be a promising candidate for transdifferentiation of fibroblasts to neurons.

  1. Lines 258 – 263 – I would delete the general sentence about crosslinking and I would rephrase the sentence starting from line 259 like:  Glutaraldehyde (GA) is a well-known crosslinker….

“Crosslinking with chemical substances is widely used in polymerization process” Was deleted from the manuscript.

  1. The summary is to specific, it is not covering all the chapters presented in this review. The hydrogels are mentioned only ones. Please complete the summary at least with one phrase about the findings from every chapter.

6. Summary

Fibroblasts can be used for tissue regeneration in the future. By manipulation of epigenetic and transcription factor systems, fibroblasts present the possibility to be reprogrammed for the generation of iPSCs and transdifferentiationinto neurons. Biomimetic surfaces such as hydrogels could potentially be a valuable tool for generation of iPSCs and transdifferentiation of somatic cells. Interaction forces between cells and surfaces pass through the nucleus which works as a mechanosensor. Tensile force and shear stress can be applied to hydrogels which act as the transporter of these stimuli to cells. Nucleus deformations can influence chromatin conformation that might directly modulate gene expression by altering the nature of the protein complexes resulting in modified global patterns of chromatin histone acetylation and transcriptional regulation of gene expression. The major concern about reprogramming is the adverse effects. Using epigenetic modifiers can put stresses on cells during reprogramming which can lead to the mutation of stress regulatory genes which prone to tumor formation. Instead of trying to force a cell with epigenetic modifiers, transfected transcription factors or chromatin remodeling drugs, it may be safe to try more natural tools, like biomimetic biomaterials which are similar to what occurs during natural reprogramming and transdifferentiation [89].If it’s necessary, further manipulation such as increased expression of particular transcription factors could be used as a parallel technique to help permit functional transdifferentiation. Incorporation of chemical or biological substances can improve the properties of hydrogels to become more bioactive. Combination of various polymer types can modulate the desirable elasticity. We end by stating that we recognize that materials directed transdifferentiation is a moot point, despite having been described in major papers [57,90]. We run with these major observations and focus on the potential of fibroblasts and hydrogels to help neural regenerative strategies.

  1. Please recheck the hole manuscript for typos (ex: line 259 substanc)

I have tried to correct the typos error, if it still occurs, please suggest.

Reviewer 2 Report

I would like to support this revised review paper (Manuscript number processes-1127749-peer-review-v2) for publication in Processes. All suggested changes were made (or discussed/clarified) by the authors. The discussion is informative and clear. To summarize, I think that this paper can be published as-is.

Author Response

Dear Reviewer

Thank you very much.